# OpenReview forum: "BaxBench: Can LLMs Generate Correct and Secure Backends?"
_ICLR.cc/2025/Workshop/BuildingTrust — BuildingTrust_

### Official Review · Reviewer_apoQ · 2025-02-19

**Rating:** 7
**Confidence:** 4

**Review:**

This paper introduces BAXBENCH, a novel benchmark designed to evaluate the capabilities of LLMs in generating accurate and secure backend applications. BAXBENCH combines 28 scenarios and 14 frameworks and evaluates 10 state - of - the - art LLMs using functional tests and security exploits. The findings of this paper offer a valuable benchmark and insightful experimental results.

**Questions:**

1. Given that the paper's experiments used general models, would there be performance discrepancies for code-domain fine-tuned models when subjected to the BAXBENCH benchmark proposed herein
2. Can the single - run experiment results precisely represent the models' performance stability?
3. Are there any schemes or approaches to dynamically update the security exploit code library to keep pace with the constantly changing security environment?

---

### Official Review · Reviewer_T8y8 · 2025-03-01
**Evaluating LLMs for Secure Backend Code Generation**

**Rating:** 6
**Confidence:** 3

**Review:**

Summary

The paper presents a benchmark for evaluating LLMs' capability in application backend code generation. It includes 28 scenarios, each describing an application requirement, API specification, environment-specific instructions, and database requirements. The generated code is evaluated through functional tests and security assessments.

For security evaluations, they manually craft common exploit code targeting specific CWEs for each scenario. Their results show that current state-of-the-art (SOTA) models still struggle with correctness, achieving only around 60% accuracy. Furthermore, nearly half of the code that passes functional tests contains vulnerabilities. They also demonstrate that prompting with security hints can help reduce vulnerabilities but does not fully mitigate the issue.

Strength

A very useful benchmark for evaluating the LLM's capability on backend code across 14 very popular backend framework. Security is a big issue as more practitioner using LLM to quickly generate backend code from scratch. This work can raise awareness on the security vulnerability issue when developers use LLM tools without aware of the security risk.

Weakness

it would be interesting to also test LLM agent's performance on the benchmark as many practioners' nowadays are using agent to develop.

---

### Decision · Program_Chairs · 2025-03-04

Accept